# Fluorine-Containing, Self-Assembled Graft Copolymer for Tuning the Hydrophilicity and Antifouling Properties of PVDF Ultrafiltration Membranes

**DOI:** 10.3390/polym15173623

**Published:** 2023-09-01

**Authors:** Seung Jae Moon, Young Jun Kim, Du Ru Kang, So Youn Lee, Jong Hak Kim

**Affiliations:** Department of Chemical and Biomolecular Engineering, Yonsei University, Seoul 03722, Republic of Korea

**Keywords:** surface modification, ultrafiltration membrane, PVDF, graft copolymer, water permeance

## Abstract

Neat poly(vinylidene fluoride) (PVDF) ultrafiltration (UF) membranes exhibit poor water permeance and surface hydrophobicity, resulting in poor antifouling properties. Herein, we report the synthesis of a fluorine-containing amphiphilic graft copolymer, poly(2,2,2-trifluoroethyl methacrylate)-*g*-poly(ethylene glycol) behenyl ether methacrylate (PTFEMA-*g*-PEGBEM), hereafter referred to as PTF, and its effect on the structure, morphology, and properties of PVDF membranes. The PTF graft copolymer formed a self-assembled nanostructure with a size of 7–8 nm, benefiting from its amphiphilic nature and microphase separation ability. During the nonsolvent-induced phase separation (NIPS) process, the hydrophilic PEGBEM chains were preferentially oriented towards the membrane surface, whereas the superhydrophobic PTFEMA chains were confined in the hydrophobic PVDF matrix. The PTF graft copolymer not only increased the pore size and porosity but also significantly improved the surface hydrophilicity, flux recovery ratio (FRR), and antifouling properties of the membrane. The membrane performance was optimal at 5 wt.% PTF loading, with a water permeance of 45 L m^−2^ h^−1^ bar^−1^, a BSA rejection of 98.6%, and an FRR of 83.0%, which were much greater than those of the neat PVDF membrane. Notably, the tensile strength of the membrane reached 6.34 MPa, which indicated much better mechanical properties than those reported in the literature. These results highlight the effectiveness of surface modification via the rational design of polymer additives and the precise adjustment of the components for preparing membranes with high performance and excellent mechanical properties.

## 1. Introduction

Ultrafiltration (UF) membranes have attracted considerable attention because of their ability to separate clean water from various materials such as oils [1,2,3], wastewater [4,5,6,7], proteins [8,9], and nanoparticles [10]. Among the various polymers used for UF membranes, poly(vinylidene fluoride) (PVDF) [11,12,13], polysulfone (PSf) [14,15], and poly(ether sulfone) (PES) [16,17,18] are commonly employed because of their cost-effectiveness, high mechanical strength, and ease of preparation. However, the hydrophobic nature of these membranes still poses a challenge, as it leads to membrane fouling by hydrophobic biomolecules and organic matter, limiting the effective application of these membranes in biotechnology and water treatment [19].

Surface modification and blending techniques have been investigated as a means of enhancing the hydrophilicity and antifouling properties of UF membranes. Surface modification techniques include surface coating [20,21] and graft polymerization [22,23]. However, these methods often result in decreased water permeance because the surface pores of the membranes are easily blocked. Alternatively, blending copolymers as additives during membrane preparation yields membranes with controlled pores and high porosity [19,24,25,26]. This approach allows for the efficient rejection of foulants without sacrificing the water permeance. For a successful membrane preparation, the polymer additives should exhibit good compatibility with the host matrix and impart hydrophilicity to the membrane surface, resulting in enhanced antifouling properties [13,25,27].

In this study, a fluorine-containing amphiphilic graft copolymer was synthesized via a facile free-radical polymerization method to meet the aforementioned requirements. This graft copolymer, henceforth referred to as PTF, comprises superhydrophobic poly(2,2,2-trifluoroethyl methacrylate) (PTFEMA) and hydrophilic poly(ethylene glycol) behenyl ether methacrylate (PEGBEM). The PTF graft copolymer was blended into a PVDF matrix to prepare porous UF membranes via nonsolvent-induced phase separation (NIPS). Polyethylene glycol (PEG), a representative pore-forming agent, was used in the preparation process to further enhance the membrane porosity. The successful synthesis of the PTF graft copolymer was confirmed by Fourier-transform infrared (FTIR) spectroscopy and proton nuclear magnetic resonance (^1^H NMR). The morphology and nanostructure of the PTF graft copolymer were characterized using transmission electron microscopy (TEM), X-ray diffraction (XRD), and small-angle X-ray scattering (SAXS). The interactions, morphologies, and microstructures of the resultant UF membranes were investigated using FTIR spectroscopy, scanning electron microscopy (SEM), XRD, and permeation analyses. The hydrophilicity of the membranes was examined by contact angle (CA) and flux recovery ratio (FRR) measurements.

## 2. Materials and Methods

### 2.1. Materials

2,2,2-Trifluoroethyl methacrylate (TFEMA, 99%), poly(ethylene glycol) behenyl ether methacrylate solution (PEGBEM, *M_n_* ≈ 1500 g mol^−1^, 50 wt.% in methacrylic acid/water), poly(vinylidene fluoride) (PVDF, *M_w_* ≈ 534,000 g mol^−1^), poly(ethylene glycol) (PEG, *M_n_* = 400 g mol^−1^), and bovine serum albumin (BSA, pH 7, ≥98%) were purchased from Sigma-Aldrich (St. Louis, MO, USA). 2,2′-Azobis(2-methylpropionitrile) (AIBN, 98%) was purchased from Acros Organics (Geel, Belgium) and used as the initiator for free-radical polymerization. All chemicals were used as received without further purification.

### 2.2. Synthesis of the PTF Graft Copolymer

The PTF graft copolymer was synthesized via a conventional free-radical polymerization. First, 18 g of PEGBEM was dissolved in 90 mL of ethyl acetate (EtOAc) in a round-bottom flask under stirring at 25 °C. Subsequently, 2 g of TFEMA and 0.1 g of AIBN initiator were added to the solution. After purging with N_2_ gas for 30 min, the solution was immersed in an oil bath at 70 °C and stirred for 18 h. Then, the reaction mixture was precipitated using excess *n*-hexane. This washing process was performed thrice, and the resulting copolymer was completely dried in a vacuum oven at 50 °C for 12 h.

### 2.3. Preparation of UF Membranes

A series of PVDF UF membranes was prepared using the NIPS method. The membrane casting solution was prepared in 25 mL vials based on the composition in Table 1. First, four different amounts of PVDF were dissolved by immersion in a solvent mixture of tetrahydrofuran (THF)/dimethylformamide (DMF) (1:1, *v*/*v*) at 25 °C for 24 h. Subsequently, 0.15 g of PEG pore-forming agent and specific amounts of the PTF graft copolymer were added to the solution in sequence, followed by vigorous stirring for 24 h. The total solute concentration in the solution was maintained at 18% (*w*/*v*). After complete dissolution of the solids, the obtained solution was degassed without stirring for 2 h until the gas bubbles completely disappeared. The homogeneous solution was then cast to a thickness of 100 μm on a flat glass plate using a doctor blade and immersed in a 25 °C coagulation bath filled with deionized (DI) water. After peeling off the glass plate, the prepared membranes were soaked in DI water for at least 48 h before the UF permeation tests to completely wash out any PEG remaining in the membranes. The relative humidity and temperature during membrane preparation were fixed at 75% and 25 °C, respectively.

### 2.4. Characterization of the PTF Graft Copolymer

The synthesized PTF graft copolymer was characterized by Fourier-transform infrared spectroscopy (FTIR, Spectrum 100, Perkin Elmer, Waltham, MA, USA) in the frequency range of 2500–600 cm^−1^. The composition of the copolymer was characterized by a 400 MHz ^1^H NMR spectrometer (AVANCE III HD 400, Bruker Biospin, Ettlingen, Germany) with CD_3_COCD_3_ as the solvent. The molecular weights of the copolymers were determined by gel permeation chromatography (GPC, Young Lin Instrument, Seoul, Republic of Korea). The nanoscale structure of the copolymer was observed using a transmission electron microscope (TEM; JEM-F200, JEOL, Tokyo, Japan) at an acceleration voltage of 200 kV. To prepare the TEM sample, a 1 wt.% polymer solution in THF was directly cast onto a TEM grid, and the solvent was completely removed by evaporation in a vacuum oven at 50 °C for 12 h. Small-angle X-ray scattering (SAXS) analysis was conducted using the 4C SAXS II beamline at the Pohang Accelerator Laboratory, Republic of Korea.

### 2.5. Characterization of the PVDF UF Membranes

#### 2.5.1. Morphology and Microstructure

The surface and cross-sectional morphologies of the PVDF membranes were observed using field-emission scanning electron microscopy (FE-SEM) (JEOL-7610F-Plus, JEOL, Tokyo, Japan). Each membrane was immersed in liquid nitrogen and broken into cross sections. All the membranes were coated with a thin layer of platinum prior to observation. The mean pore size of the membranes was determined using a permeabilizer (POROLUX™ 1000, IB-FT GmbH, Berlin, Germany). The overall porosities of the UF membranes were calculated using the following equation:(1)$$\varepsilon \left(\%\right)=\frac{w_{wet}-w_{dry}}{\rho_{w}\times A\times h}\times 100$$
where *ε* (%) is the porosity of the membrane, *w_wet_* and *w_dry_* (g) are the weights of the wet membrane and dried membrane, *ρ_w_* (g cm^−3^) is the density of DI water, *A* (cm^2^) is the effective area, and *h* (cm) is the thickness of the membrane.

#### 2.5.2. Hydrophilicity of the Membranes’ Surface

The contact angle between water and the PVDF UF membranes was measured to evaluate the hydrophilicity of the membranes’ surface using an optical tensiometer (TL100, Biolin Scientific, Espoo, Finland). The contact angles were measured at five random points and averaged.

#### 2.5.3. Permeation and Separation Performance

The permeance of pure water and a BSA solution (1 g L^−1^) was measured using a flat sheet membrane test system (PHILOS, Gwangmyeong-si, Republic of Korea) at 25 °C and a pressure of 1 bar (Appendix A). The active area of the membrane was 12.56 cm^2^. First, the pure water permeance was recorded over the course of 1 h at 1 bar. Subsequently, the feed tank was filled with the BSA solution, and the permeance was measured over 1 h under the same pressure. The fouled membrane was thoroughly cleaned by washing with DI water for 10 min, and the measurement was repeated over 1 h to determine the recovered water permeance. This cycle was repeated twice. An identical membrane was prepared and tested four times, and the average value was recorded as the permeation and separation performance of the membrane. The concentrations of BSA in the feed and permeate were measured by UV–vis spectroscopy (Hewlett-Packard, Palo Alto, CA, USA) at 278 nm (Appendix A). The permeance (*P*) and rejection rates (*R*) were calculated as follows:(2)$$P=\frac{∆V}{A\times ∆t\times p}$$
(3)$$R=\left(1-\frac{C_{p}}{C_{f}}\right)\times 100$$
where ∆*V* is the volume of the permeated water (L), *A* is the active area (m^2^), ∆*t* is the permeation time (h), and *p* is the operating pressure (bar); *C_p_* and *C_f_* are the concentrations (mg L^−1^) of BSA in the permeate and feed solutions, respectively.

#### 2.5.4. Antifouling Performance

The flux recovery ratio (FRR) was calculated to investigate the antifouling performance of the membranes using the following equation:(4)$$FRR=\left(\frac{P_{w2}}{P_{w1}}\right)\times 100$$
where *P_w_*_1_ is the initial pure water permeance, and *P_w_*_2_ is the recovered pure water permeance in the second cycle.

## 3. Results and Discussion

### 3.1. Synthesis of the PTF Graft Copolymer

To design a polymeric additive suitable for surface modification of PVDF UF membranes, an amphiphilic PTF graft copolymer was synthesized using the TFEMA monomer and PEGBEM macromonomer via a one-step free-radical polymerization process using AIBN as an initiator (Figure 1). A series of PTF graft copolymers with various TFEMA/PEGBEM ratios was synthesized, but only the PTF copolymer with a TFEMA/PEGBEM ratio of 1:9 was investigated in this study. This is because the PTF copolymers with lower TFEMA contents (<10%) are soluble in water, whereas those with higher TFEMA contents (>10%) show poor chain mobility (too glassy) and thus inefficient alignment with the surface, resulting in deteriorated water permeance and antifouling properties. The synthesized PTF graft copolymer was in the shape of a white solid chunk with robust mechanical strength. The yield was as high as about 95%, indicating a highly favorable reaction mechanism. The PEGBEM chains consist of 25 ether and 20 methyl groups, which endow the synthesized PTF graft copolymer with hydrophilic and semi-crystalline properties, respectively. Moreover, the superhydrophobic CF_3_ groups of the TFEMA chain exhibit good compatibility with the PVDF matrix, owing to the chemical similarity and polarity of the two species. Therefore, it was expected that the PTF graft copolymer containing the hydrophobic PTFEMA and hydrophilic PEGBEM domains would function as an effective agent for enhancing the surface hydrophilicity of the membrane by strengthening the attachment of the PTFEMA chains to the PVDF matrix and aligning the PEGBEM chains toward the membrane surface.

FTIR spectra were obtained to confirm the successful synthesis of the PTF graft copolymer, as shown in Figure 1a. The TFEMA monomer and PEGBEM macromonomer exhibited weak absorption bands at 1639 and 1632 cm^−1^, respectively, which were assigned to the C=C stretching vibration mode. These absorption bands completely disappeared in the spectrum of the PTF graft copolymer, confirming that the polymer synthesis via free-radical polymerization was successful, and no unreacted residual monomers were present. The absorption bands of the PTF graft copolymer at 1742 and 1704 cm^−1^ were assigned to the stretching vibration of C=O bonds, which resulted from the TFEMA monomer (original band at 1734 cm^−1^) and the PEGBEM macromonomer (original band at 1697 cm^−1^), respectively [28,29,30]. A strong absorption band appeared at 1274 cm^−1^ in the spectrum of the TFEMA monomer, attributed to the stretching vibration of the CF_3_ groups, which shifted to 1280 cm^−1^ in the spectrum of the PTF graft copolymer [30]. The absorption bands at 1108 and 1063 cm^−1^, assigned to the stretching vibration mode of the C-O-C groups, were also observed for the PTF graft copolymer. These absorption bands resulted from the C-O-C groups of the TFEMA monomer at 1136 cm^−1^ and of the PEGBEM macromonomer at 1088 cm^−1^, which were significantly red-shifted by 28 and 25 cm^−1^, respectively [30]. It was deduced that specific bonding interactions, such as hydrogen bonding and dipole–dipole interactions, were strengthened after graft polymerization. The characteristic absorption bands of the TFEMA monomer and the PEGBEM macromonomer were present after graft copolymerization, indicating that both PTFEMA and PEGBEM chains were well incorporated in the resulting PTF graft copolymer.

The exact composition of the PTF graft copolymer was determined using ^1^H NMR spectroscopy (Figure 1b) by integrating the corresponding chemical shift of each species, i.e., -CH_2_- protons from the TFEMA monomer at 4.6 ppm (denoted “a”) and -CH_2_- protons from the PEGBEM macromonomer at 1.3 ppm (denoted “b”) [29,30]. The copolymer composition matched the feeding ratio of the two monomers. The actual molar ratio was 61.8:38.2 (TFEMA/PEGBEM), and the corresponding weight ratio was 15.5:84.5. The actual weight ratio of the TFEMA chains was slightly higher than the initial feed weight ratio (1:9), because the much larger volume of the PEGBEM macromonomer with respect to that of the TFEMA monomer resulted in greater steric hindrance during the propagation step. For GPC analysis, THF was used as the solvent. The weight-average molecular weight (*M_w_*) of the PTF graft copolymer was approximately 63,000 g mol^−1^, with a dispersity (*M_w_*/*M_n_*) of 1.3.

### 3.2. Structure and Morphology of the PTF Graft Copolymer

The morphology of the PTF graft copolymer was characterized using TEM, as shown in Figure 2. The PTF graft copolymer had a self-assembled, microphase-separated nanostructure on the size scale of 7–8 nm, attributed to its amphiphilic nature [30,31,32]. The dark region in the Figure corresponds to the hydrophobic PTFEMA domains owing to the presence of fluorine, which is a relatively heavy atom with a high electron density. In contrast, the bright region represents the hydrophilic PEGBEM domains. The microphase separation of the polymer facilitated the preferential orientation of the hydrophilic PEGBEM chains towards the surface of the PVDF membranes during the phase inversion process, whereas the superhydrophobic PTFEMA moieties remained in the interior. This selective arrangement enhanced the surface hydrophilicity of the membranes, leading to improved water affinity and enhanced antifouling properties.

The structure and crystallinity of the PTF graft copolymers were investigated using XRD, as shown in Figure 3a. The XRD pattern of the poly(TFEMA) homopolymer synthesized exclusively from the TFEMA monomer displayed a single broad peak centered at 2θ = 18.0°, indicating its amorphous nature. In contrast, the XRD profile of the poly(PEGBEM) homopolymer synthesized solely from the PEGBEM macromonomer showed a broad amorphous peak as well as two additional sharp peaks of crystalline species at 2θ angles of 19.2° and 23.2°, corresponding to the diffractions of the (120) and (112) planes of the semi-crystalline poly(ethylene oxide) chains, respectively [33]. After graft copolymerization, the sharp peaks of the crystalline species were barely observed, indicating the amorphous nature of the resulting PTF graft copolymer. This indicated that graft copolymerization was a facile and effective approach for reducing the degree of crystallinity of the polymer, which contributed to the surface segregation of the PTF graft copolymer due to the increased chain mobility.

The nanoscale structure of the PTF graft copolymer was characterized using SAXS, as shown in Figure 3b. By applying the Bragg’s relation (*d* = 2π/*q*), the *d*-spacing value of the PTF graft copolymer was determined to be 8.1 nm, with a corresponding *q* value of 0.77 nm^−1^. The value obtained by SAXS was consistent with the polymer chain distance observed in the TEM images. These findings provided further supporting evidence of the self-assembled nanostructures of the PTF graft copolymers.

### 3.3. Preparation and Properties of PVDF/PTF Membranes

The PTF graft copolymer was carefully designed to contain superhydrophobic PTFEMA and hydrophilic PEGBEM chains. The CF_3_ groups in the TFEMA chains may plausibly enhance the compatibility with the hydrophobic PVDF matrix via dipole–dipole interactions, while the ethylene oxide groups in the PEGBEM chains increase the surface hydrophilicity of the membrane by aligning towards the membrane surface. First, the interaction between the PVDF matrix and the PTF graft copolymer additive was investigated using FTIR spectroscopy, as shown in Figure 4a. PVDF exhibits complex crystal polymorphism, consisting of three main phases: α, β, and γ [34,35]. The chains in α-phase PVDF have zero dipole moments, whereas the β-phase has the largest number of polar chains. Strong absorption bands at 873 and 1066 cm^−1^ were observed for the neat PVDF, resulting from the presence of the α-phase PVDF. The other two absorption bands at 841 and 1182 cm^−1^ correspond to β-phase PVDF [34]. Interestingly, the two bands of the α-phase were, respectively, blue-shifted by +2 and +6 cm^−1^ after adding the PTF graft copolymer, whereas the bands of the β-phase were red-shifted by −2 and −5 cm^−1^. The blue shift indicated a stronger bonding force of the α-phase itself, implying a repulsive force between the PTF copolymer and the α-phase of PVDF. On the other hand, the red shift indicated a decreased bonding force of the β-phase itself, implying an attraction force between the PTF copolymer and the β-phase PVDF. This result suggested that the PTF graft copolymer preferentially attracted the β-phase of PVDF, forming secondary bonding interactions with the polar moiety of PVDF. Therefore, the PTF graft copolymer provides the advantage of establishing water permeation pathways by endowing the PVDF matrix with a more phase-separated structure.

XRD was used to investigate the changes in the crystallinity of the PVDF membranes upon incorporation of the PTF graft copolymer (Figure 4b). Owing to the predominance of α-phase PVDF, distinct diffraction peaks were observed in the XRD pattern. These crystalline peaks corresponded to the (020), (110), and (021) reflections of the monoclinic α-phase crystal at 2θ angles of 18.3°, 19.9°, and 26.6°, respectively [34,36]. Interestingly, upon incorporation of the PTF graft copolymer, the intensities of all three crystalline peaks increased, and the width of the peak centered at 19.9° became significantly narrower. This indicates an increase in the degree of crystallinity of the PVDF matrix, which was consistent with the FTIR spectroscopy data, indicating repulsive interactions between the PTF graft copolymer and the α-phase of the PVDF matrix.

High-performance UF membranes for practical separation processes must have excellent mechanical properties, such as high tensile strength, sufficient flexibility, and appropriate elasticity [36]. The tensile properties of the PVDF/PTF membranes were evaluated, as shown in Appendix A. It is well known that the incorporation of polymer additives generally leads to a decrease in the tensile strength and elongation of PVDF membranes [37,38,39]. This trend was observed for the PVDF/PTF membranes and was attributed to the differences in the chemical structures of the PVDF matrix and the PTF graft copolymer. Despite the decrease in the tensile strength with PTF loading, the PVDF/PTF3 and PVDF/PTF5 membranes exhibited a remarkably high tensile strength exceeding 6 MPa, which is one of the highest values reported for PVDF-based UF membranes [37,38,39]. These findings indicated that the PVDF/PTF membranes maintained excellent mechanical strength for practical applications, making them a promising option for use in various membrane processes.

To investigate the effect of the PTF graft copolymer on the membrane morphology, surface and cross-sectional images of the PVDF UF membranes were obtained using FE-SEM, as shown in Figure 5. The PVDF/PTF0 membrane exhibited a rather dense surface with a low pore density, even after the extraction of PEG (pore-forming agent) with water (Figure 5a). When 3 wt.% of the PTF graft copolymer was incorporated, a large number of surface pores was generated, because the PTF graft copolymer possesses much higher hydrophilicity than the PVDF matrix, allowing it to rapidly align on the membrane surface during the phase separation process [19]. As the content of the PTF copolymer increased, larger pores were obtained. However, the surface porosity decreased when the weight percentage of the PTF copolymer exceeded 3 wt.% (Appendix A). This could be due to the abundance of hydrophilic components, resulting in the formation of larger pores during phase separation and a decrease in the pore density on the membrane surface. The cross-sectional images of the membranes (Figure 5b,d,f,h) showed some larger pores (macrovoids) near the surface, resulting from the alignment of the hydrophilic copolymer. As the content of the PTF graft copolymer increased, there was a noticeable development of macrovoids. Consequently, the overall porosity increased linearly from 61% (PVDF/PTF0) to 71% (PVDF/PTF10), as shown in Appendix A. The PVDF/PTF3, 5, and 10 membranes were also approximately 10–15% thicker than the PVDF membrane without PTF. The bottom side of all membranes displayed a sponge-like morphology owing to the slow phase separation caused by the lower affinity of THF for water compared to other organic solvents, such as DMF.

The interconnected active pores of the PVDF/PTF membranes were characterized by permeabilization, as shown in Figure 6. The results were obtained by analyzing the relationship between the N_2_ gas flow rates and pressure applied across the membrane while the pores were saturated with the wetting liquid (Porefil) [40,41]. The *Y*-axis, labeled as the percentage flow, represents the percentage of the corresponding pore diameter, and the sum of each bar adds up to 100%. The PVDF/PTF0 membrane, which was prepared without the PTF copolymer, was excluded from the measurements as it was water-impermeable. The PVDF/PTF3 membrane exhibited the narrowest pore size distribution, which was suitable for the efficient rejection of the target molecules. The mean flow pore size gradually increased with the increasing PTF copolymer content, indicating that higher copolymer loadings resulted in the formation of larger pores (Table 2). In addition, a higher content of the PTF graft copolymer led to a wider pore size distribution. Therefore, it can be concluded that the optimal pore size for the rejection of BSA was achieved with the incorporation of 3–5 wt.% PTF graft copolymer.

### 3.4. Permeation and Antifouling Properties of the PVDF/PTF Membranes

The surface hydrophilicity has a significant effect on the water permeation and antifouling properties of PVDF membranes. The surface hydrophilicity can be evaluated by measuring the contact angle (CA) of a water droplet on the membrane surface, which reflects the affinity of the membrane for water molecules [42,43]. A lower CA value indicates greater hydrophilicity of the membrane surface. The CA values of the PVDF UF membranes with various PTF loadings were examined; the results are shown in Figure 7. As the content of the PTF copolymer increased to 10 wt.%, the CA declined significantly from 66.8° to 55.5°. This suggested that the surface hydrophilicity of the PVDF membrane was significantly enhanced by incorporating the PTF graft copolymer and that the membrane hydrophilicity could be easily adjusted by varying the copolymer content. Notably, however, the CA of the PVDF/PTF10 membrane was slightly lower than that of the PVDF/PTF5 membrane, which indicated that the efficiency of surface hydrophilization decreased after a certain amount of PTF graft copolymer was added, which is similar to the results reported in other studies [9,44,45]. This can be attributed to the saturation of the surface hydrophilization mechanism, according to which the hydrophobic PTFEMA domain is oriented towards the PVDF matrix side, and the hydrophilic PEGBEM chains are concentrated on the surface. When 5 wt.% of PTF graft copolymer was added, the distribution of the hydrophilic PEGBEM chains on the surface reached the maximum concentration, resulting in effective surface modification.

The permeance of membranes for pure water and the BSA solutions is influenced by several factors, such as pore size, porosity, pore structure, and surface hydrophilicity. As confirmed previously, the addition of the PTF graft copolymer directly affected these factors. Consequently, the changes in the permeation and antifouling properties of the PVDF membranes with the addition of the PTF graft copolymer were investigated (Figure 8a and Table 3). Notably, the PVDF/PTF0 membrane, which had a dense surface and was impermeable to water, was excluded from the analysis. Upon adding 3 wt.% PTF graft copolymer, the membrane became water-permeable due to the formation of surface pores. When the content of the PTF copolymer was increased to 10 wt.%, the pure water permeance of the membrane increased linearly from 44.7 to 54.1 L m^−2^ h^−1^ bar^−1^, which represented an increase of approximately 20%. This suggested that incorporation of the PTF graft copolymer led to the development of well-connected water transport channels within the PVDF membranes. Although the surface hydrophilicity of the PVDF/PTF10 membrane was comparable to that of the PVDF/PTF5 membrane, the water permeance of the former was higher. This can be attributed to the larger mean pore size of the PVDF/PTF10 membrane, which was approximately 32% greater than that of PVDF/PTF5 (Table 2). As a result, there was a significant decrease in BSA rejection for the PVDF/PTF10 membrane, as shown in Figure 8b.

The adsorption of rejected proteins on the membrane surface during membrane operation results in a significant reduction in the membrane permeance. Hence, antifouling is a crucial membrane property that can be assessed by determining the FRR value [46,47]. The FRR is primarily influenced by the surface hydrophilicity of the membrane. A more hydrophilic surface establishes stronger interactions with water molecules, allowing the facile removal of hydrophobic foulants from the surface. The PVDF/PTF5 membrane had a significantly higher FRR value (83%) than the PVDF/PTF3 membrane, which was consistent with the CA data presented in Figure 7. This indicated that the surface hydrophilicity of the membranes was correlated with their antifouling performance. However, the FRR value decreased to 76.4% for the PVDF/PTF10 membrane. This may be attributed to the fact that among the tested membranes, the PVDF/PTF10 membrane had the largest mean pore size, which resulted in the lowest BSA rejection. A lower BSA rejection generally leads to more pronounced membrane fouling and increases the likelihood of residual BSA adhering to the membrane even after washing. Therefore, despite having a slightly lower CA than the PVDF/PTF5 membrane, the PVDF/PTF10 membrane exhibited a lower FRR. Overall, the water permeance of the PVDF/PTF5 membrane was slightly higher, and the BSA rejection was slightly lower than those of the PVDF/PTF3 membrane. However, owing to its significantly higher surface hydrophilicity, PVDF/PTF5 exhibited the highest FRR value among all the tested membranes. Based on these results, we concluded that the PVDF/PTF5 membrane was the optimal sample we examined in terms of overall performance, achieving a good balance between water permeance, BSA rejection, and antifouling properties. Consequently, PVDF membranes containing the PTF graft copolymer additive offer the advantage of being easily cleaned with water alone, eliminating the requirement for harsh chemical cleaning procedures. This suggests that the operational lifetimes of these membranes can be prolonged, ultimately reducing the membrane process costs.

A comparison between the PVDF/PTF3 and the PVDF/PTF5 membranes was carried out to demonstrate the improved membrane performance. The increase in water flux and FRR was calculated and subsequently compared with those for other membranes reported in the literature, as summarized in Table 4. The increase in the modifier content was calculated relative to the PVDF mass and not to the total mass. The relative increase in the flux was calculated using the second water permeance data. Even when the PTF content increased only to 2.6 wt.%, the FRR value increased remarkably, indicating the superior ability of the PTF graft copolymer for enhanced surface hydrophilicity of the membrane. The observed increase in the FRR values surpassed those observed for other PVDF UF membranes reported in the literature. This highlighted the effectiveness of the PTF graft copolymer as an efficient modifier for enhancing the performance of PVDF UF membranes.

The PVDF/PTF membranes exhibited excellent mechanical properties, surpassing those of other PVDF UF membranes reported in the literature (Figure 9 and Table 5). The outstanding mechanical strength of the PVDF/PTF membranes is attributed to the specific interaction and good miscibility of the PTF graft copolymer with the PVDF matrix, without macrophase separation between the two macromolecules. Furthermore, the robust properties of the PTF graft copolymer derive from its high molecular weight (i.e., 63,000 g mol^−1^), amphiphilic nature, and chemical structure containing long repeated methyl and ethylene oxide groups that could contribute to the excellent mechanical strength of the membranes. These physical properties make the PVDF/PTF membranes an advantageous choice for practical membrane applications that require durability and reliability.

## 4. Conclusions

In this study, we proposed the synthesis and utilization of a fluorine-containing self-assembled PTF graft copolymer as a surface modifier for enhancing the porosity and hydrophilicity of PVDF UF membranes. The PTF graft copolymer, synthesized through free-radical polymerization, exhibited a unique self-assembled microphase-separated nanostructure with a size of 7–8 nm, due to its amphiphilic properties. By aligning the hydrophilic PEGBEM chains towards the membrane surface, the pore structure and surface hydrophilicity of PVDF membranes could be tuned. Increasing the content of the PTF graft copolymer gradually increased both the pore size and the overall porosity, resulting in higher water permeances. Incorporating 10 wt.% of the PTF graft copolymer caused a significant decrease in BSA rejection, indicating a threshold concentration of the PTF graft copolymer. The surface hydrophilicity of the membranes was significantly enhanced after incorporation of the PTF graft copolymer, resulting in increased FRR values and enhanced antifouling properties. Overall, the PVDF/PTF5 membrane with 5 wt.% loading emerged as the optimal choice, exhibiting a water permeance of 45 L m^−2^ h^−1^ bar^−1^, a BSA rejection of 98.6%, and the highest FRR value of 83.0%. Importantly, the tensile strength of the PVDF/PTF5 membrane was as high as 6.34 MPa, which is four times greater than the values reported in the literature. The modification of PVDF membranes with a small amount of PTF graft copolymer proved to be an economical and effective approach for preparing membranes with high separation performance and excellent mechanical strength.

## Data Availability

Not applicable.

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
