# Peer review of "Fluorine-Containing, Self-Assembled Graft Copolymer for Tuning the Hydrophilicity and Antifouling Properties of PVDF Ultrafiltration Membranes"

_polymers, 2023, doi:10.3390/polym15173623_

Round 1
Reviewer 1 Report
The authors have prepared a a fluorine-containing amphiphilic graft copolymer. This copolymer was blended with PVDF at different weight ratio for preparaing the ultrafiltration membrane. Various properties associated with these UF membranes were examined. Further, the authors compared the UF membrane prepared in this study with those prepared in the literatures. This contribution is well presented and the conclusions are supported by the results presented.
Author Response
Response
Thank you very much for your positive reviews and comments.

Author Response
Reviewer #1
The manuscript described the synthesis of a novel amphiphilic polymer PTF containing fluorinated and PEG groups. The dipole-dipole interaction provided strong affinity between PTF and PVDF, while the amphiphilicity of PTF induced self-assembled microphase-separation during the membrane fabrication process. The generated membrane was well characterized by surface analysis and permeability test. This membrane showed outstanding permeability and meanwhile a good BSA rejection. The properties were controllable by adjusting PTF amount during membrane fabrication. This manuscript was well-organized and demonstrated the application of a novel polymer. However, some experimental and discussion details are needed.
Comments #1
- Line 194 and Figure 1b: The peak at 4.6 ppm was barely visible at the current scale. Providing an inset showing the magnified peak A region is better.
Response #1
As suggested, Figure 1b has been revised with an inset.
Comments #2
- For polymer characterization, please provide more information including yield, conversion, appearance (color, powder or other states), and the solvent for GPC.
Response #2
As suggested, the information on the yield (line 163), appearance (line 161), and solvent for GPC (line 204) has been added as follows.
“The synthesized PTF graft copolymer was in the shape of a white solid chunk with robust mechanical strength. The yield was as high as about 95%, indicating a highly favorable reaction mechanism.”
“For GPC analysis, THF was used as the solvent.”
Comments #3
- The dash lines in Figure 4 and Figure 8a are not clear enough.
Response #3
As suggested, Figure 4 and Figure 8a have been revised with clear dash lines.
Comments #4
- Line 253-255: Please provide some references showing why the red-shifts for β-phase peaks and blue-shifts for α-phase peaks indicated that PTF only interacted with β-phase.
Response #4
We meant that the PTF graft copolymer attracts the beta-phase rather than the alpha-phase. We have revised the unclear sentence and added a detailed description (line 259-264) as follows.
“The blue-shift indicates a stronger bonding force of the α-phase itself, implying a repulsive force between the PTF copolymer and the α-phase of PVDF. On the other hand, the red-shift indicates a decreased bonding force of the β-phase itself, implying an attraction force between the PTF copolymer and the β-phase PVDF. This result suggests that the PTF graft copolymer preferentially attracts the β-phase of PVDF”
Comments #5
- Standard deviations were needed for several measurements, including Figure S2, Table 2 and Table 3, Table 4 and Figure 8b to show the data reproducibility and better support the discussion, such as the wider pore size distribution claimed by the author in Line 322-323.
Response #5
The structure (such as pore size distribution and porosity) and separation performance of membranes are strongly dependent on the membrane preparation conditions such as humidity and temperature. Thus, the membrane preparation condition was added (line 95-96)
“The relative humidity and temperature for membrane preparation were fixed at 75% and 25 °C, respectively.”
For the data reproducibility, we have added the following description (line 136-138)
“The identical membrane was prepared and tested four times, and the average value was recorded as the permeation and separation performance of the membrane.”
Comments #6
- The permeability was described in 2 units: LMH and LMH/bar (e.g., Figure 8a). Though the values are the same as the pressure in this work was constant at 1 bar, it’s better to use only one unit through the manuscript to avoid misunderstanding.
Response #6
As suggested, L m-2 h-1 bar-1 was chosen as the permeation unit throughout the manuscript.
Comments #7
- Please provide a detailed description of the permeability and filtration test. Adding a scheme of the setup is helpful. Is it dead-end or cross-flow filtration? Different filtration types may significantly affect the rejection and fouling results, so the comparison of membrane performance using different filtration types may be unfair. For example, data from reference 9 in Table 5 is based on cross-flow filtration, while data from reference 37 is based on dead-end filtration.
Response #7
A cross-flow filtration system was used for our study, and the corresponding schematic diagram has been included in the Supplementary Information (Figure S1). In Table 4 and 5, we have added a column to indicate if the filtration type is dead-end or cross-flow.
Comments #8
- Please provide a UV/Vis spectrum of BSA in the supporting information.
Response #8
As suggested, UV-vis spectra of BSA solutions were presented in Figure S2.
